# State Representation Learning Using an Unbalanced Atlas

**Li Meng**
University of Oslo
Oslo, Norway
li.meng@its.uio.no

**Morten Goodwin**
University of Agder
Kristiansand, Norway
morten.goodwin@uia.no

**Anis Yazidi**
Oslo Metropolitan University
Oslo, Norway
anisy@oslomet.no

**Paal Engelstad**
University of Oslo
Oslo, Norway
paal.engelstad@its.uio.no

## Abstract

The manifold hypothesis posits that high-dimensional data often lies on a lower-dimensional manifold and that utilizing this manifold as the target space yields more efficient representations. While numerous traditional manifold-based techniques exist for dimensionality reduction, their application in self-supervised learning has witnessed slow progress. The recent MSimCLR method combines manifold encoding with SimCLR but requires extremely low target encoding dimensions to outperform SimCLR, limiting its applicability. This paper introduces a novel learning paradigm using an unbalanced atlas (UA), capable of surpassing state-of-the-art self-supervised learning approaches[1]. We investigated and engineered the DeepInfomax with an unbalanced atlas (DIM-UA) method by adapting the Spatiotemporal DeepInfomax (ST-DIM) framework to align with our proposed UA paradigm. The efficacy of DIM-UA is demonstrated through training and evaluation on the Atari Annotated RAM Interface (AtariARI) benchmark, a modified version of the Atari 2600 framework that produces annotated image samples for representation learning. The UA paradigm improves existing algorithms significantly as the number of target encoding dimensions grows. For instance, the mean F1 score averaged over categories of DIM-UA is ∼75% compared to ∼70% of ST-DIM when using 16384 hidden units.

## 1 Introduction

Self-supervised learning (SSL) is a field in machine learning (ML) that aims to learn useful feature representations from unlabelled input data. SSL includes mainly contrastive methods (Oord et al., 2018; Chen et al., 2020; He et al., 2020) and generative models (Kingma & Welling, 2013; Gregor et al., 2015; Oh et al., 2015). Generative models rely on using generative decoding and reconstruction loss, whereas typical contrastive methods do not involve a decoder but apply contrastive similarity metrics to hidden embeddings instead (Liu et al., 2021).

State representation learning (SRL) (Anand et al., 2019; Jonschkowski & Brock, 2015; Lesort et al., 2018) focuses on learning representations from data typically collected in a reinforcement learning (RL) environment. A collection of images can be sampled through an agent interacting with the environment according to a specified behavior policy. Such images are interesting as study subjects due to their innate temporal/spatial correlations. Moreover, RL can also benefit from self-supervised learning just as computer vision (CV) and natural language processing (NLP) do, and successful pretraining of neural network (NN) models may lead to improvements in downstream RL tasks.

A manifold can be learned by finding an atlas that accurately describes the local structure in each chart (Pitelis et al., 2013). In SSL, using an atlas can be viewed as a generalization of both dimen-

---

[1]Code is available at https://github.com/mengli11235/DIM-UA.

sionality reduction and clustering (Korman, 2018; 2021a;b). Namely, it generalizes the case where only one chart exists and where the charts do not overlap in an atlas. In MSimCLR (Korman, 2021b), NNs can encode an atlas of a manifold by having chart embeddings and membership probabilities.

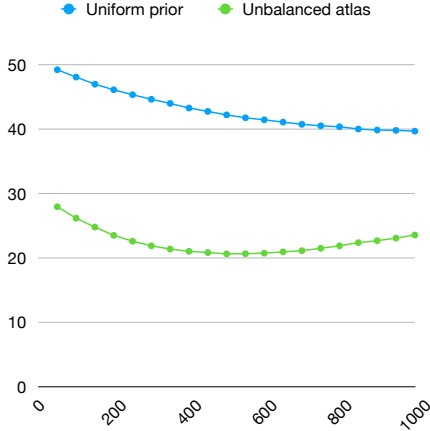

Figure 1: The entropy of the output vector recorded epoch-wise when pretrained on the CIFAR10 dataset for a total of 1000 epochs, utilizing 8 charts and a dimensionality of 256.

One primary issue of MSimCLR is its reliance on a uniform prior, which allocates inputs into each chart embedding uniformly. We postulate that although this uniform prior may more effectively represent the data distribution when $d$ is exceedingly small, it concurrently introduces higher prediction uncertainty. Simultaneously, it also suffers from a problem akin to that faced by bootstrapped methods in RL. It has been noted that multiple NN heads inside a model, in the absence of additional noise, tend to output similar results after being trained a large number of epochs (Osband & Van Roy, 2015; Osband et al., 2016; Ecoffet et al., 2019; Meng et al., 2022).

To rectify the aforementioned problems, this study introduces a novel SSL paradigm that leverages an unbalanced atlas (UA). In this context, UA denotes the absence of a uniform prior distribution, with the membership probability distribution deliberately trained to deviate significantly from uniformity. As illustrated in Fig. 1, it is evident that the entropy of the output vector during pretraining when using UA is markedly lower than that with a uniform prior, which suggests a heightened degree of confidence in its predictions.

Our contribution is summarized as follows: (1) We modify the SRL algorithm ST-DIM (Anand et al., 2019) with our UA paradigm and introduce a new algorithm called DIM-UA. This furthers the research into the integration of RL and SSL with a novel manifold-based learning paradigm. DIM-UA achieves the state-of-the-art performance on samples collected from 19 Atari games of the AtariARI benchmark. (2) We also provide detailed ablations and additional experiments on CIFAR10 to examine different underlying effects of possible design choices. (3) We demonstrate that our UA paradigm is capable of effectively representing a manifold with a large number (e.g., $\geq 256$) of hidden dimensions, whereas previous research (Korman, 2021a;b) only showed promise with a small number (e.g., $\leq 8$) of hidden dimensions. The UA paradigm thereby showcases its capability to build larger models, transcending the constraints imposed by model backbones.

## 2 RELATED WORK

**Dimensionality reduction with manifolds**  It is common for nonlinear dimensionality reduction (NLDR) algorithms to approach their goals based on the manifold hypothesis. For example, the manifold structure of an isometric embedding can be discovered by solving for eigenvectors of the matrix of graph distances (Tenenbaum et al., 2000). A sparse matrix can also be used instead with a locally linear embedding (Roweis & Saul, 2000). Correspondence between samples in different data sets can be recovered through the shared representations of the manifold (Ham et al., 2003). Manifold regularization provides an out-of-sample extension compared to graph-based approaches

(Belkin et al., 2006). Manifold sculpting simulates surface tension progressively in local neighborhoods to discover manifolds (Gashler et al., 2007).

**Self-supervised learning** There are relevant works on generative models, such as variational autoencoders (VAEs) (Kingma & Welling, 2013) and adversarial autoencoders (AAEs) (Makhzani et al., 2015). Meanwhile, contrastive methods have shown promise in the field of SSL. Contrastive Predictive Coding (CPC) learns predictive representations based on the usefulness of the information in predicting future samples (Oord et al., 2018). SimCLR provides a simple yet effective framework using data augmentations (Chen et al., 2020). Momentum Contrast (MoCo) utilizes a dynamic dictionary, which can be much larger than the mini-batch size (He et al., 2020). The recent trend within research in contrastive learning has been on removing the need for negative pairs. BYOL utilizes a momentum encoder to prevent the model from collapsing due to a lack of negative pairs (Grill et al., 2020). SimSiam further shows that a stop-gradient operation alone is sufficient (Chen & He, 2021). Barlow Twins, on the other hand, achieves so by minimizing the redundancy of vector components outputted by two identical networks that take distorted versions of inputs (Zbontar et al., 2021).

**Self-supervised learning with manifolds** Representing non-Euclidean data in NN models is a key topic in geometric deep learning (Bronstein et al., 2017). Learning manifolds using NNs was explored in (Korman, 2018), in which AAEs were used to learn an atlas as latent parameters. Constant-curvature Riemannian manifolds (CCMs) of different curvatures can be learned similarly using AAEs (Grattarola et al., 2019). Mixture models of VAEs can be used to express the charts and their inverses to solve inverse problems (Alberti et al., 2023). A combination of autoencoders and Barlow Twins can capture both the linear and nonlinear solution manifolds (Kadeethum et al., 2022).

## 3 METHOD

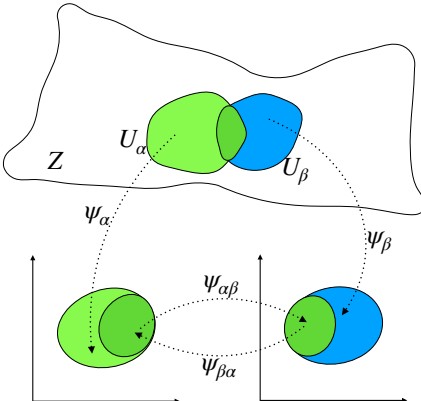

Figure 2: A manifold $Z$ embedded in a higher dimension. Two domains are denoted by $U_\alpha$ and $U_\beta$ in $Z$. $\psi_\alpha$ and $\psi_\beta$ are the corresponding charts that map them to a lower dimensional Euclidean space. An atlas is then a collection of these charts that together cover the entire manifold.

Our method extends the work from Korman (2021b) and builds a manifold representation using multiple output embeddings and membership probabilities of those embeddings. First, an illustration of a manifold $Z$ is in Fig. 2. Instead of directly learning an encoder of $Z$, we can learn encoder functions of $\psi_\alpha(U_\alpha)$ and $\psi_\beta(U_\beta)$ together with a score function. For a given input, we can specify which encoder to use according to the output of the score function.

We formally model a distribution of input data by a manifold as follows: $x$ is the input from input space $\mathcal{X}$, $\mathcal{Z}$ is the latent space, $f$ is an embedding function: $\mathcal{X} \to \mathcal{Z}$. $\mathcal{I}$ is the identity mapping, $d$ the number of dimensions for each chart output embedding, $n$ the number of charts, and $\mathcal{N}$ denotes $\{1, 2, ..., n\}$. $\psi_i \colon \mathcal{Z} \to \mathbb{R}^d$ is the inverse mapping of a coordinate map: $\mathbb{R}^d \to \mathcal{Z}$, whereas $q = (q_1, q_2, ..., q_n) \colon \mathcal{Z} \to [0, 1]^n$ is the chart membership function. The output of our model is then given by Eq. 1.

$$\text{Output}(x) = \sum q_i(f(x))\mathcal{I}(\psi_i(f(x))) \tag{1}$$

At inference time, the one-hot encoding of $q(x)$ is used instead (Eq. 2).

$$\text{Output}(x) = \mathcal{I}(\psi_i(f(x))), \text{ where } i = \text{argmax}_j \, q_j(f(x)) \tag{2}$$

## 3.1 UNBALANCED ATLAS

Like other SSL methods with manifolds (Korman, 2021b;a), UA uses a maximal mean discrepancy (MMD) objective (Gretton et al., 2012; Tolstikhin et al., 2017), which is defined by Eq. 3.

$$\text{MMD}_k(P_1, P_2) = \|\int_{\mathcal{S}} k(s, \cdot)dP_1(s) - \int_{\mathcal{S}} k(s, \cdot)dP_2(s)\|_{\mathcal{H}_k} \tag{3}$$

Here, $k$ is a reproducing kernel, $\mathcal{H}_k$ is the reproducing kernel Hilbert space of real-valued functions mapping $\mathcal{S}$ to $\mathbb{R}$, and $P_1$, $P_2$ are distributions on the space.

In our paradigm, the input $x$ is designed to be represented in charts with higher membership probabilities. Thus, we take an MMD loss that moves the conditional membership distribution far away from the uniform distribution. We use the kernel $k_{\mathcal{N}} : \mathcal{N} \times \mathcal{N} \to \mathbb{R}$, $(i, j) \to \delta_{ij}$, and $\delta_{ij} = 1$ if $i = j$ else 0, and thus have Eq. 4.

$$\mathcal{L}_{\mathcal{N}}(q) = -\mathbb{E}_z \text{MMD}_{k_{\mathcal{N}}}(q(z), \mathcal{U}_{\mathcal{N}}) = -\mathbb{E}_z \sum_{i=1}^{n} (q_i(z) - \frac{1}{n})^2 \tag{4}$$

Here, $\mathcal{U}_{\mathcal{N}}$ denotes the uniform distribution on $\mathcal{N}$, $z$ is the embedding of $f(x)$.

Unlike MSimCLR, we do not use an MMD objective to make the prior distribution to be uniform, but take another approach to improve the model stability and head diversity when $d$ is not trivial. In Fig. 2, $U_\alpha \cap U_\beta$ has transitive maps in their respective chart coordinates with domains restricted to $\psi_\alpha(U_\alpha \cap U_\beta)$ and $\psi_\beta(U_\alpha \cap U_\beta)$, which are $\psi_{\alpha\beta} = \psi_\beta \circ \psi_\alpha^{-1}$ and $\psi_{\beta\alpha} = \psi_\alpha \circ \psi_\beta^{-1}$. What interests us the most is this intersection between $U_\alpha$ and $U_\beta$. More precisely, since the overlapping representations in each head of the model become a dominant negative factor when $d$ grows larger, we aim at modelling a manifold with dilated prediction targets in pretraining to avoid convergent head embeddings and collapsing solutions. We use the average values of chart outputs to model a Minkowski sum (Mamatov & Nuritdinov, 2020; Wang et al., 2020), which serves a key purpose in our paradigm.

While convergent head embeddings should be avoided, the learning process should not break the convergence entirely. Proposition 1 implies that the Minkowski sum of the output embeddings contains the Minkowski sum of all mappings of intersections, which means that using dilated prediction targets by taking the Minkowski sum does not omit any mapped intersected embedding.

**Proposition 1.** *Let $U = \{U_1, U_2, ..., U_n\}$ be a collection of open subsets of $Z$ whose union is all of $Z$, and $\bigcap_{i=1}^{n} U_i$ is not empty. For each $i \in \{1, 2, ..., n\}$, there is a homeomorphism $\psi_i : U_i \to V_i$ to an open set $V_i \subset \mathbb{R}^d$. We have the Minkowski sum $V_i + V_j = \{a + b \mid a \in V_i, b \in V_j\}$. Then $\sum_{i=1}^{n} \psi_i(\bigcap_{j=1}^{n} U_j) \subset \sum_{i=1}^{n} V_i$.*

*Proof.* For any vector $a \in \sum_{i=1}^{n} \psi_i(\bigcap_{j=1}^{n} U_j)$, there exists $a_i \in \psi_i(\bigcap_{j=1}^{n} U_j)$ such that $a = \sum_{i=1}^{n} a_i$. Because $\psi_i(\bigcap_{j=1}^{n} U_j) \subset V_i$, we also have $a_i \in V_i$, $i \in \{1, 2, ..., n\}$. Then $\sum_{i=1}^{n} a_i \in \sum_{i=1}^{n} V_i$, $a \in \sum_{i=1}^{n} V_i$ and thus $\sum_{i=1}^{n} \psi_i(\bigcap_{j=1}^{n} U_j) \subset \sum_{i=1}^{n} V_i$.

□

Proposition 2 further states that the average Minkowski sum of the output embeddings together with the Minkowski sum of the average of mappings of intersections can be used instead and still keeps Proposition 1 true, under the assumption that each mapping of the intersection is convex. More generally, Proposition 1 holds true with scalar multiplications when convexity is assumed. However, it should be noted that convexity is not guaranteed here. In Eq. 1, we approach this assumption by using an identity mapping $\mathcal{I}$ instead of a linear mapping from Korman (2021b). More about the convexity assumption is addressed in Appendix B.

**Proposition 2.** *Let $U = \{U_1, U_2, ..., U_n\}$ be a collection of open subsets of $Z$ whose union is all of $Z$, and $\bigcap_{i=1}^{n} U_i$ is not empty. For each $i \in \{1, 2, ..., n\}$, there is a homeomorphism $\psi_i : U_i \to V_i$ to an open set $V_i \subset \mathbb{R}^d$. The multiplication of set $V$ and a scalar $\lambda$ is defined to be $\lambda V = \{\lambda a \mid a \in V\}$. We take the Minkowski sum. If each $\psi_i(\bigcap_{j=1}^{n} U_j)$ is convex, then $\sum_{i=1}^{n} \frac{1}{n} \psi_i(\bigcap_{j=1}^{n} U_j) \subset \frac{1}{n} \sum_{i=1}^{n} V_i$.*

*Proof.* Follows Proposition 1 and the property of scalar multiplication, $\frac{1}{n} \sum_{i=1}^{n} \psi_i(\bigcap_{j=1}^{n} U_j) \subset \frac{1}{n} \sum_{i=1}^{n} V_i$. Since scalar multiplication is preserved for convex sets, we have $\sum_{i=1}^{n} \frac{1}{n} \psi_i(\bigcap_{j=1}^{n} U_j) \subset \frac{1}{n} \sum_{i=1}^{n} V_i$.

□

## 3.2 DIM-UA

We experiment with our UA paradigm using the SRL algorithm ST-DIM (Anand et al., 2019), and propose DIM-UA. ST-DIM develops on Deep InfoMax (DIM)(Hjelm et al., 2018) that uses infoNCE (Oord et al., 2018) as the mutual information estimator between patches. Its objective consists of two components. One is the global-local objective ($\mathcal{L}_{GL}$) and the other one is the local-local objective ($\mathcal{L}_{LL}$), defined by Eq. 5 and Eq. 6 respectively.

$$\mathcal{L}_{GL} = \sum_{m=1}^{M} \sum_{n=1}^{N} -\log \frac{\exp(g_{m,n}(x_t, x_{t+1}))}{\sum_{x_{t*} \in X_{next}} \exp(g_{m,n}(x_t, x_{t*}))} \tag{5}$$

$$\mathcal{L}_{LL} = \sum_{m=1}^{M} \sum_{n=1}^{N} -\log \frac{\exp(h_{m,n}(x_t, x_{t+1}))}{\sum_{x_{t*} \in X_{next}} \exp(h_{m,n}(x_t, x_{t*}))} \tag{6}$$

Here, $x_t$ and $x_{t+1}$ are temporally adjacent observations, whereas $X_{next}$ is the set of next observations and $x_{t*}$ is randomly sampled from the minibatch. $M$ and $N$ are the height and width of local feature representations.

We denote the encoder as $f$, the output (global feature) vector of input $x_t$ as Output$(x_t)$, and the local feature vector of $x_t$ at point $(m, n)$ as $f_{m,n}(x_t)$. $W_g$ and $W_h$ are linear layers that will be discarded in probing. Then, we have the score function $g_{m,n}(x_t, x_{t+1}) = \text{Output}(x_t)^T W_g f_{m,n}(x_{t+1})$, and $h_{m,n}(x_t, x_{t+1}) = f_{m,n}(x_t)^T W_h f_{m,n}(x_{t+1})$ of ST-DIM.

For DIM-UA, we need to redefine the score function of $\mathcal{L}_{GL}$ by Eq. 7 because the UA paradigm utilizes dilated prediction targets during pretraining, where $\psi_i(f(x_t))$ is the output of $x_t$ from the $i$-th head following encoder $f$ for each $i$ in $\mathcal{N}$.

$$g_{m,n}(x_t, x_{t+1}) = [\frac{1}{n} \sum_{i=i}^{n} \psi_i(f(x_t))]^T W_g f_{m,n}(x_{t+1}) \tag{7}$$

According to Eq. 4 , we have the MMD objective $\mathcal{L}_Q$ defined as Eq. 8, where $q_i(f(x_t))$ is the membership probability of the $i$-th head for each $i$ in $\mathcal{N}$ when the input is $x_t$.

$$\mathcal{L}_Q = -\frac{1}{2}\sum_{i=1}^{n}((q_i(f(x_t)) - \frac{1}{n})^2 + (q_i(f(x_{t+1})) - \frac{1}{n})^2) \tag{8}$$

Thereby, the UA objective (Eq. 9) is a sum of above objectives, where $\tau$ is a hyper-parameter.

$$\mathcal{L}_{UA} = \mathcal{L}_{GL} + \mathcal{L}_{LL} + \tau\mathcal{L}_Q \tag{9}$$

## 4   EXPERIMENTAL DETAILS

The performance of DIM-UA and other SRL methods is evaluated on 19 games of the AtariARI benchmark. There are five categories of state variables in AtariARI (Anand et al., 2019), which are agent localization (Agent Loc.), small object localization (Small Loc.), other localization (Other Loc.), miscellaneous (Misc.), and score/clock/lives/display (Score/.../Display).

We follow the customary SSL pipeline and record the probe accuracy and F1 scores on the downstream linear probing tasks. The encoder is first pretrained with SSL, and then is used to predict the ground truth of an image with an additional linear classifier. Notably, the weights of the encoder are trained only during the pretraining and are fixed in the probing tasks. The data for pretraining and probing are collected by an RL agent running a certain number of steps using a random policy since it was found that the samples collected by a random policy could be more favorable than those collected by policy gradient policies for SSL methods (Anand et al., 2019).

Previous SSL methods in Anand et al. (2019) have used a single output head with 256 hidden units. One of the major interests in our experiment is to discover the effect of choosing different values for the number of dimensions $d$ and the number of charts $n$. Therefore, we scale up the number of hidden units, while keeping the model architecture, to observe the performance of using a single output head without UA and of using multiple heads with UA. To make a fair comparison, we compare the performance when the total number of hidden units are equal, i.e., when $1 \times d$ for a single output head and $n \times d$ for multiple output heads are equal in our AtariARI experiment. In contrast, we also modify SimCLR using our UA paradigm and follow the customs from MSimCLR to compare the performance of different methods with $d$ being equal (Korman, 2021b) in additional experiments on CIFAR10.

The experiments are conducted on a single Nvidia GeForce RTX 2080 Ti and 8-core CPU, using PyTorch-1.7 (Paszke et al., 2019). An illustration of the model backbone, hyper-parameters, and pseudocode of the algorithm are accompanied in Appendix A.

## 5   RESULTS

In this section, we show the empirical results of our experiments and compare DIM-UA with other SSL methods to verify the efficacy of our UA paradigm. Meanwhile, we pay special attention to the performance of models when choosing different values for $n$ and $d$.

For a straightforward comparison, we first observe the probe F1 scores together with standard deviations of each game averaged across categories in Table 1. "ST-DIM*" denotes ST-DIM with one output head of 16384 units, while DIM-UA uses four output heads with 4096 units in each head. We compare them to various methods of using a single output head of 256 hidden units here, which are taken from Anand et al. (2019). Each table entry is an average of 5 independent pretraining/probing runs using images sampled from different seeds. The probe accuracy scores are also included in Appendix A.

Using 16384 units ("ST-DIM*") does not necessarily guarantee better performance than using 256 units (ST-DIM). In 7 out of 19 games, "ST-DIM*" has lower F1 scores than ST-DIM. In particular, the model collapses due to overfitting when using too many units to represent the global features on Freeway. As a result, "ST-DIM*" only gains an F1 score of 0.3 with standard deviation 0.355 on Freeway. The mean F1 score of "ST-DIM*" is only 0.7 compared to 0.72 of ST-DIM. On the other

Table 1: Probe F1 scores of each game averaged across categories

| Game | VAE | CPC | ST-DIM | ST-DIM* | DIM-UA |
|---|---|---|---|---|---|
| Asteroids | 0.36 | 0.42 | 0.49 | $0.48 \pm 0.005$ | $\mathbf{0.5} \pm 0.007$ |
| Bowling | 0.50 | 0.90 | **0.96** | $\mathbf{0.96} \pm 0.021$ | $\mathbf{0.96} \pm 0.018$ |
| Boxing | 0.20 | 0.29 | 0.58 | $0.61 \pm 0.008$ | $\mathbf{0.64} \pm 0.007$ |
| Breakout | 0.57 | 0.74 | 0.88 | $0.88 \pm 0.02$ | $\mathbf{0.9} \pm 0.016$ |
| Demon Attack | 0.26 | 0.57 | 0.69 | $0.71 \pm 0.01$ | $\mathbf{0.74} \pm 0.012$ |
| Freeway | 0.01 | 0.47 | 0.81 | $0.3 \pm 0.355$ | $\mathbf{0.86} \pm 0.02$ |
| Frostbite | 0.51 | **0.76** | 0.75 | $0.73 \pm 0.005$ | $0.75 \pm 0.004$ |
| Hero | 0.69 | 0.90 | 0.93 | $0.93 \pm 0.008$ | $\mathbf{0.94} \pm 0.004$ |
| Montezuma Revenge | 0.38 | 0.75 | 0.78 | $0.81 \pm 0.016$ | $\mathbf{0.84} \pm 0.014$ |
| Ms Pacman | 0.56 | 0.65 | 0.72 | $0.74 \pm 0.017$ | $\mathbf{0.76} \pm 0.011$ |
| Pitfall | 0.35 | 0.46 | 0.60 | $0.69 \pm 0.031$ | $\mathbf{0.73} \pm 0.029$ |
| Pong | 0.09 | 0.71 | 0.81 | $0.78 \pm 0.015$ | $\mathbf{0.85} \pm 0.004$ |
| Private Eye | 0.71 | 0.81 | 0.91 | $0.91 \pm 0.009$ | $\mathbf{0.93} \pm 0.009$ |
| Qbert | 0.49 | 0.65 | 0.73 | $0.78 \pm 0.026$ | $\mathbf{0.79} \pm 0.02$ |
| Seaquest | 0.56 | 0.66 | 0.67 | $0.68 \pm 0.007$ | $\mathbf{0.69} \pm 0.007$ |
| Space Invaders | 0.52 | 0.54 | 0.57 | $0.59 \pm 0.007$ | $\mathbf{0.62} \pm 0.013$ |
| Tennis | 0.29 | 0.60 | 0.60 | $0.57 \pm 0.018$ | $\mathbf{0.64} \pm 0.025$ |
| Venture | 0.38 | 0.51 | **0.58** | $0.57 \pm 0.014$ | $\mathbf{0.58} \pm 0.01$ |
| Video Pinball | 0.45 | 0.58 | 0.61 | $0.6 \pm 0.031$ | $\mathbf{0.62} \pm 0.023$ |
| Mean | 0.41 | 0.63 | 0.72 | $0.7 \pm 0.033$ | $\mathbf{0.75} \pm 0.013$ |

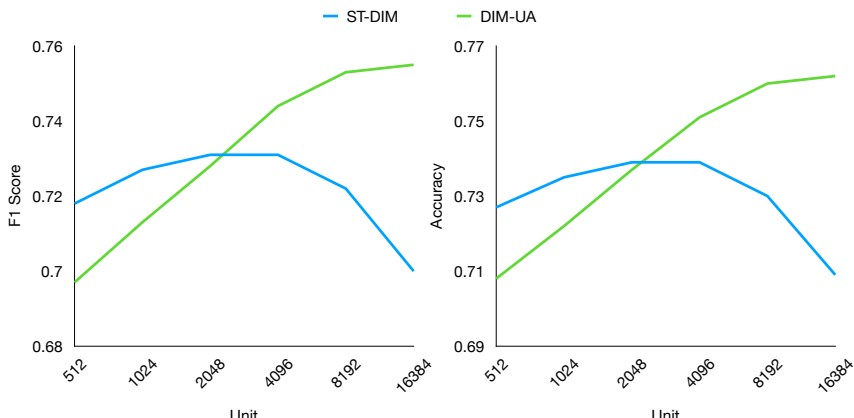

Figure 3: The mean F1 and accuracy scores of 19 games when the total number of hidden units varies. The number of heads for DIM-UA is set to 4 here.

hand, DIM-UA achieves higher scores and more stable performance. The F1 scores of DIM-UA are equal or higher than those of both "ST-DIM*" and ST-DIM in every game. The mean F1 score of DIM-UA is 0.75, the highest among all methods in Table 1.

## 5.1 ABLATIONS

In this subsection, our goal is to observe the behavior of models with different settings while the total number of units ($n \times d$) on the horizontal axis of figures varies. Fig. 3 examines the difference between ST-DIM and DIM-UA. Fig. 4 examines the effects of changing the number of heads for DIM-UA. In addition, we compare DIM-UA with two methods designed on ST-DIM in Fig. 5. One method uses the paradigm from MSimCLR, denoted by "+MMD". The other is similar to ours, which minimizes the loss of Eq. 9, but without modifying the score function as in Eq. 7 (namely, DIM-UA without using dilated prediction targets), denoted by "-UA". The 6 games mentioned here are Asteroids, Breakout, Montezuma Revenge, Private Eye, Seaquest, and Video Pinball.

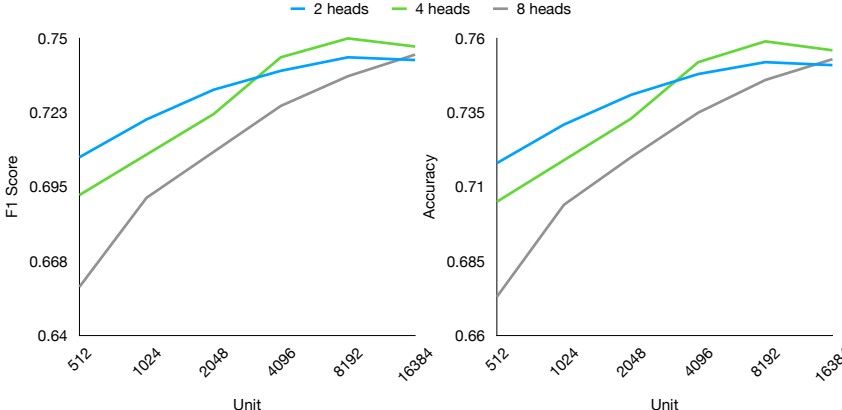

Figure 4: The mean F1 and accuracy scores of DIM-UA on 6 games when the number of output heads is 2, 4, or 8.

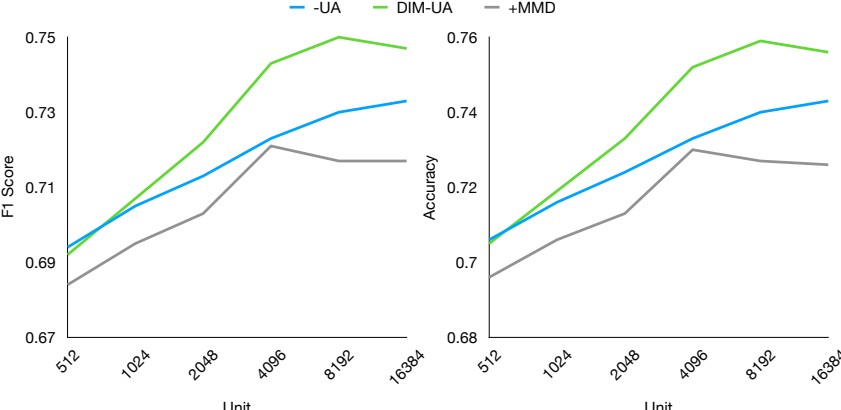

Figure 5: The mean F1 and accuracy scores on 6 games with different adaptations. All methods use 4 output heads.

In Fig. 3, ST-DIM performs better than DIM-UA when the number of hidden units is small. Their scores become close to each other when the number of units is around 2048. DIM-UA continues to improve as the total number of units grows, whereas the performance of ST-DIM drops at the same time. It is expected for DIM-UA to have lower F1 and accuracy scores when the encoding dimensions are low since the diversity among output heads demands more epochs of training. However, the efficacy of our UA paradigm is clearly demonstrated, as it allows the model to extend its capability by continuously expanding the encoding dimensions.

Since we expect DIM-UA to converge slower because of the diversity among output heads, we expect this to become more obvious as the number of heads increases. We can verify that in Fig. 4, where the model with two output heads has the highest F1 and accuracy scores when the total number of hidden units is below 2048. On the other hand, it obtains the lowest F1 and accuracy score when the total number of units grows to 16384. Meanwhile, the model with eight output heads gives the worst results when the number of units is small but shows no sign of plateau, even with very high encoding dimensions. Increasing $n$ while keeping $d$ the same in our UA paradigm helps with the manifold representation but also lowers the performance if $d$ is not large enough.

In Fig. 5, it is not surprising that "+MMD" obtains the worst results in spite of the number of units, since MSimCLR was only found out to be helpful when the number of units is extremely small (e.g., 2, 4). "-UA" obtains better results than DIM-UA when the number of units is 512 but gets overrun by DIM-UA when the number becomes even larger. This empirically demonstrates that the dilated prediction targets in our UA paradigm are critical to achieve effective manifold representations.

Table 2: Linear evaluation accuracy on CIFAR10

| Method | Head | Dimension | | |
|---|---|---|---|---|
| | | 256 | 512 | 1024 |
| SimCLR | - | $0.881 \pm 0.002$ | $0.883 \pm 0.002$ | $0.881 \pm 0.003$ |
| MSimCLR | 2 | $0.877 \pm 0.002$ | $0.878 \pm 0.001$ | $0.866 \pm 0.003$ |
| MSimCLR | 4 | $0.873 \pm 0.001$ | $0.873 \pm 0.001$ | $0.861 \pm 0.002$ |
| MSimCLR | 8 | $0.864 \pm 0.001$ | $0.859 \pm 0.005$ | $0.857 \pm 0.002$ |
| SimCLR-UA | 2 | $0.882 \pm 0.001$ | $0.884 \pm 0.001$ | $0.885 \pm 0.001$ |
| SimCLR-UA | 4 | $0.885 \pm 0.001$ | $0.884 \pm {<}0.001$ | $0.88 \pm 0.001$ |
| SimCLR-UA | 8 | $0.882 \pm {<}0.001$ | $\mathbf{0.886} \pm 0.002$ | $0.876 \pm 0.005$ |

## 5.2 Additional Experiments on CIFAR10

We modify SimCLR using the UA paradigm (SimCLR-UA) and perform additional experiments on CIFAR10, following the parameter settings and evaluation protocol from Korman (2021b); Chen et al. (2020). SimCLR-UA uses multiple heads with dilated prediction targets instead in pretraining, and adds $\mathcal{L}_Q$ in Eq. 8 to the contrastive loss of SimCLR. Here, ResNet50 is used as the backbone, which is significantly larger than the default backbone in ST-DIM. We also slightly modify our model here based on an ablation study on CIFAR10. Please check Appendix B for more details.

In Table 2, each entry is an average accuracy score obtained from three independent pretraining and evaluation runs, together with the standard deviation. SimCLR obtains an accuracy score of 88.3% when it uses 512 hidden units. In the mean time, SimCLR-UA achieves the highest accuracy score of 88.6% among three methods when it uses eight heads with 512 units in each. The second best score is also achieved by SimCLR-UA, which is 88.5% when using four heads with 256 units in each head, or using two heads with 1024 units in each. We acknowledge that our improvement over SimCLR is small under this experimental setup. Nonetheless, there is a significant increase in accuracy when comparing SimCLR-UA to MSimCLR, especially in the case where the number of heads is larger. For instance, the highest evaluation accuracy score of MSimCLR is 87.8%, 87.3% and 86.4% respectively, when using two, four or eight heads. In contrast, SimCLR-UA obtains the highest accuracy score when using eight heads. This supports our hypothesis that UA can be a universal paradigm to effectively create manifold representations in SSL.

## 6 Discussion

We have demonstrated that our UA paradigm helps improve the performance of both ST-DIM and SimCLR when encoding dimensions are high. Furthermore, we argue that training NNs with multiple output heads is inherently slower and more demanding than training with a single output head, which has restrained the study in its domain. It is evident that our paradigm can overcome this headwind by generating effective manifold representations. Moreover, our UA paradigm has gained a significant amount of improvement compared to the most related state-of-the-art manifold representation paradigm MSimCLR in the experiments on AtariARI and CIFAR10.

Notably, the UA paradigm also exhibits the potential of modeling a manifold using further higher dimensions while increasing the number of output heads (Fig. 4). It can be an important contribution because this means the performance of the model scales with the size of output heads. Using 16384 hidden units in total is not very efficient economically when the entire model is small, but the additional overhead introduced by doing so can be relatively insignificant when the model itself is large. In particular, this trade-off may also be worthwhile in challenging downstream tasks where the smallest increase in probe accuracy can make a difference.

Our work has illustrated that SSL methods with manifolds have great potential, and more topics can be researched in this area. The relationship between the number of hidden units and the number of output heads in an NN model demands more study (see Appendix B for more discussion on this). The convexity assumption is crucial in representing the manifold. Future research may focus on representing a manifold using an unbalanced atlas more efficiently, e.g., designing new objectives and convexity constraints.

ACKNOWLEDGMENTS

This work was performed on the [ML node] resource, owned by the University of Oslo, and operated by the Department for Research Computing at USIT, the University of Oslo IT-department.

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

## A  DETAILS OF DIM-UA

Table 3 provides values of some crucial hyper-parameters experimented on AtariARI that are kept the same across different methods. In addition, $\tau$ in Eq. 9 is set to $0.1$ for DIM-UA.

Fig. 6 illustrates the standard backbone in the AtariARI experiment. The output values of the last convolutional layer in the backbone are taken as the feature map to get the local feature vector $f_{m,n}(x_t)$ of input $x_t$ at location $(m, n)$. For the original ST-DIM, a fully connected layer of 256 units immediately follows the backbone. For DIM-UA, a projection head ($\psi$) and a membership probability head ($q$) branch from the backbone.

Table 3: The values of hyper-parameters on AtariARI

| Hyper-parameter | Value |
|---|---|
| Image size | $160 \times 210$ |
| Minibatch size | 64 |
| Learning rate | 3e-4 |
| Epochs | 100 |
| Pretraining steps | 80000 |
| Probe training steps | 35000 |
| Probe testing steps | 10000 |

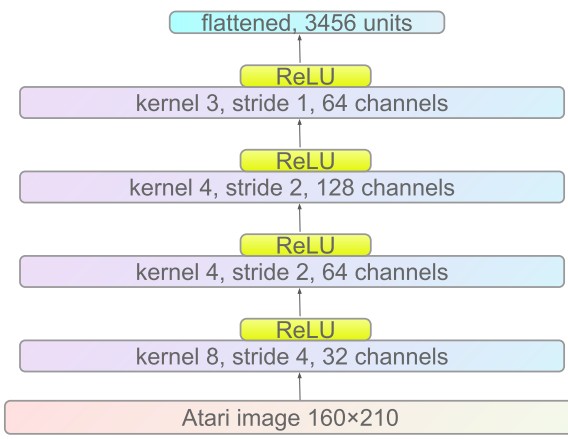

Figure 6: An illustration of the standard backbone used by ST-DIM.

Pytorch-style pseudocode of the DIM-UA algorithm is provided in Algorithm 1.

On a side note, the output of an unbalanced atlas at inference time relies on a single output head, since Eq. 4 moves the membership probability far away from the uniform distribution. As a result, the rest of the output heads does not play a role at inference time. This is different from MSimCLR, which partitions inputs into each head by simultaneously forcing a uniform prior and low entropy on conditional distributions. The role of those remaining output heads in an unbalanced atlas is comparable to the moving average network of BYOL, which produces prediction targets to help stabilize

**Algorithm 1:** Pytorch-style pseudocode for DIM-UA.

```
# f:   encoder network
# fₘ:  f but only up to the last conv layer
# pj:  projection head
# mp:  membership probability head
# c1, c2:  classifier layers only used in pretraining
#
# B: batch size
# N: number of heads
# D: number of hidden units
# C: local feature map channels
# H, W: local feature map height and width
#
# mean:  mean function along a specified dimension
# matmul:  matrix multiplication
# cross_entropy:  cross entropy loss
# mmd:  mmd loss
# size:  get the size along a specified dimension
# range:  get the range vector of an integer
# t:  transpose
#
for xₜ, xₜ₊₁ in loader:  # load B samples of xₜ, xₜ₊₁
    # get feature maps
    oₜ, yₜ, yₜ₊₁ = mean(pj(f(xₜ)), 1), fₘ(xₜ), fₘ(xₜ₊₁) # B×D, B×H×W×C
    # get the membership probabilities
    qₜ, qₜ₊₁ = mp(f(xₜ)), mp(f(xₜ₊₁)) # B×N
    # get the feature map size
    s_b, s_m, s_n = size(yₜ, 0), size(yₜ, 1), size(yₜ, 2) # B, H, W
    # initialize the loss values
    loss_g = 0, loss_l = 0
    # mmd loss
    loss = −0.05 * (mmd(qₜ) + mmd(qₜ₊₁))
    # spatial-temporal loss
    for m in range(s_m):
        for n in range(s_n):
            # global-local loss
            logits_g = matmul(c1(oₜ), yₜ₊₁[:, m, n, :].t()) # B×B
            loss_g += cross_entropy(logits_g, range(s_b)) # cross entropy loss
            # local-local loss
            logits_l = matmul(c2(yₜ[:, m, n, :]), yₜ₊₁[:, m, n, :].t()) # B×B
            loss_l += cross_entropy(logits_l, range(s_b)) # cross entropy loss
    loss_g /= (s_m * s_n), loss_l /= (s_m * s_n) # averaged by map size
    loss += loss_g + loss_l # total loss
    # optimization step
    loss.backward()
    optimizer.step()
```

the bootstrap step. The UA paradigm accomplishes a similar goal by using dilated prediction targets of output heads instead.

The probe accuracy scores are shown in Table 4, which are overall similar to the F1 scores in Table 1. The accuracy scores of DIM-UA are equal or higher than those of both "ST-DIM*" and ST-DIM in every game. The mean accuracy score of DIM-UA is 0.76, the highest among all methods.

Table 4: Probe accuracy scores of each game averaged across categories

| Game | VAE | CPC | ST-DIM | ST-DIM* | DIM-UA |
|---|---|---|---|---|---|
| Asteroids | 0.41 | 0.48 | 0.52 | $0.51 \pm 0.005$ | $\mathbf{0.53} \pm 0.007$ |
| Bowling | 0.56 | 0.90 | **0.96** | $\mathbf{0.96} \pm 0.021$ | $\mathbf{0.96} \pm 0.017$ |
| Boxing | 0.23 | 0.32 | 0.59 | $0.61 \pm 0.008$ | $\mathbf{0.64} \pm 0.007$ |
| Breakout | 0.61 | 0.75 | 0.89 | $0.89 \pm 0.019$ | $\mathbf{0.91} \pm 0.015$ |
| Demon Attack | 0.31 | 0.58 | 0.70 | $0.72 \pm 0.009$ | $\mathbf{0.74} \pm 0.011$ |
| Freeway | 0.07 | 0.49 | 0.82 | $0.33 \pm 0.34$ | $\mathbf{0.86} \pm 0.017$ |
| Frostbite | 0.54 | **0.76** | 0.75 | $0.73 \pm 0.004$ | $0.75 \pm 0.004$ |
| Hero | 0.72 | 0.90 | 0.93 | $0.93 \pm 0.008$ | $\mathbf{0.94} \pm 0.004$ |
| Montezuma Revenge | 0.41 | 0.76 | 0.78 | $0.81 \pm 0.015$ | $\mathbf{0.84} \pm 0.014$ |
| Ms Pacman | 0.60 | 0.67 | 0.73 | $0.75 \pm 0.016$ | $\mathbf{0.77} \pm 0.01$ |
| Pitfall | 0.35 | 0.49 | 0.61 | $0.7 \pm 0.028$ | $\mathbf{0.74} \pm 0.027$ |
| Pong | 0.19 | 0.73 | 0.82 | $0.79 \pm 0.014$ | $\mathbf{0.85} \pm 0.004$ |
| Private Eye | 0.72 | 0.81 | 0.91 | $0.91 \pm 0.01$ | $\mathbf{0.93} \pm 0.009$ |
| Qbert | 0.53 | 0.66 | 0.74 | $0.79 \pm 0.025$ | $\mathbf{0.8} \pm 0.019$ |
| Seaquest | 0.61 | 0.69 | 0.69 | $0.69 \pm 0.006$ | $\mathbf{0.7} \pm 0.006$ |
| Space Invaders | 0.57 | 0.57 | 0.59 | $0.6 \pm 0.009$ | $\mathbf{0.63} \pm 0.014$ |
| Tennis | 0.37 | 0.61 | 0.61 | $0.58 \pm 0.016$ | $\mathbf{0.65} \pm 0.024$ |
| Venture | 0.43 | 0.52 | **0.59** | $0.58 \pm 0.012$ | $\mathbf{0.59} \pm 0.009$ |
| Video Pinball | 0.47 | 0.59 | 0.61 | $0.6 \pm 0.03$ | $\mathbf{0.63} \pm 0.022$ |
| Mean | 0.46 | 0.65 | 0.73 | $0.71 \pm 0.031$ | $\mathbf{0.76} \pm 0.013$ |

## B  MORE ABOUT CIFAR10 EXPERIMENT

The convexity assumption is crucial in order to effectively model a manifold (e.g., scalar multiplication is not preserved when a set is non-convex). However, the universal approximation theorem implies that weights of multiple linear layers can approximate any non-convex functions. Thus, whether multiple linear layers should be used here or not could be an interesting ablation topic. Moreover, clamping can be introduced to define open sets in our method. An ablation study of SimCLR-UA is performed on CIFAR10, using 4 heads with 512 units in each head. The results are shown in Table 5, where "FC1" denotes the linear layer immediately following the ResNet50 backbone and "FC2" denotes the projection layers following coordinate mappings. The range of clamping is set to $(-10, 10)$. As the results suggest, the combination of clamping and "FC2" yields the best accuracy and is hence used to obtain results in Table 2.

Referring to the performance of SimCLR-UA in Table 2, the accuracy reaches the highest when the number of heads is eight with 512 units in each head, but when the number of heads is four, the optimal number of units is 256. It also appears that a small number of hidden units can be sufficient. This finding is different from what is observed in the AtariARI experiment, where using eight heads and 2048 units in each head is not sufficient to guarantee the optimal (Fig. 4). This may be attributed to the image size and the number of ground truth labels in an image, and more challenging tasks may demand better representations. However, we do think there should be a limited number of heads needed, related to the intrinsic dimension (ID) of data. Techniques to find ID can potentially be used to decide the optimal number of heads.

Table 5: Ablation study on CIFAR10

| FC1 | Clamp | FC2 | Accuracy |
|---|---|---|---|
| ✓ | | | 0.857 |
| | ✓ | | 0.876 |
| | | ✓ | 0.883 |
| ✓ | ✓ | | 0.872 |
| ✓ | | ✓ | 0.875 |
| | ✓ | ✓ | **0.884** |
| ✓ | ✓ | ✓ | 0.872 |

After observing the results in Table 2, it appears that the performance of SimCLR-UA could be further enhanced. Whilst comparing MSimCLR with SimCLR-UA, we maintained most hyper-parameters identical for both. However, SimCLR-UA may attain higher performance given a different set of hyper-parameters. In Fig. 1, the initial entropy of using UA is notably lower than when using a uniform prior. Ideally, it may be advantageous for the entropy to remain high during the initial stages and to decrease gradually. We suggest that the hyper-parameter $\tau$, which regulates the $\mathcal{L}_Q$ loss, could be set smaller or set to 0 initially and gradually increased over time.

We conduct an additional small-scale experiment to validate this hypothesis. In this context, we use ResNet18 as the backbone instead of ResNet50, and the training duration is set at 100 epochs as opposed to 1000. The model incorporates 8 heads, with each head containing 512 hidden units. If $\tau$ is linearly scaled, it would increment linearly, from zero up to its final value over the pretraining epochs. The outcome of this experiment is detailed in Table 6.

The table clearly illustrates that implementing linear scaling or utilizing smaller $\tau$ values can genuinely enhance the performance of SimCLR-UA. For $\tau$ values of 0.1 and 0.2, adopting a linear-scaling scheme is instrumental for optimizing the performance. However, for small $\tau$ values of 0.05 and 0.02, such a scheme is not needed. Thus, the performance of SimCLR-UA, as presented in Table 2, could potentially be boosted further, since it uses a relatively large $\tau$ value of 0.1 without any linear scaling.

Table 6: Changing $\tau$ in SimCLR-UA

| $\tau$ | Linear scaling | Accuracy |
|---|---|---|
| 0.2 |  | 0.791 |
| 0.2 | ✓ | 0.797 |
| 0.1 |  | 0.791 |
| 0.1 | ✓ | 0.8 |
| 0.05 |  | 0.799 |
| 0.05 | ✓ | 0.796 |
| 0.02 |  | **0.802** |
| 0.02 | ✓ | 0.785 |

