# OpenReview forum: "State Representation Learning Using an Unbalanced Atlas"
_ICLR.cc/2024/Conference — ICLR 2024 poster_

### Official Review · Reviewer_Ww1x · 2023-10-31

**Soundness:** 3 good
**Presentation:** 3 good
**Contribution:** 3 good
**Rating:** 8
**Confidence:** 4

**Summary:**

Authors propose to use multiple heads at the end of an encoder for
contrastive learning, instead of one. These heads are considered to
model different charts in an atlas, mapping the data manifold to the
embedding space. For a given sample, a score is computed to determine,
probabilistically, which chart should be used to encode
it. A theoretical discussion is presented to support outputting a
weighted average of charts based on scoring function, which relies on
Minkowski sum of open sets to which charts map. The scoring function
is forced to be different than a uniform distribution, thus the
unbalanced nature of the mapping. Experiments are conducted to
understand whether using multiple heads and weighted averaging at the
output leads to better representations than using a single head with
the same number of latent dimensions.

**Strengths:**

1. More expressive models for contrastive learning are very relevant
   and interesting for the community. Here authors point out that when
   the embedding dimension is very high, naive contrastive learning
   may not use the embedding space very well. Instead of using a
   single projection head, authors proposal to use multiple projection
   heads seem to lead to better results according to Tables 1
   and 2. This is a simple yet - seems to be - an effective
   modification.
2. Results of the ablation study shown in Figure 2 are very
   convincing. This simple approaches surely uses the dimensions much
   more efficiently, and provides the expected gains in accuracy.
3. The difference between +MMD and the proposed version, which I
   assume is -MMD, is striking.
4. This reviewer appreciates the experiments with CIFAR.

**Weaknesses:**

1. Technical contribution is not at a very high level, but the
   contribution is focused and pertinent.
2. In the ablation study, the model "-UA" is not clearly specified. If
   authors do not use the modifications of 7, that means the -MMD loss
   is also void. What does that yield? Are authors using a single
   projection head in this case?
3. CIFAR experiments show that the gains are much lower in these
   experiments compared to the ones obtained in ATARI games. A
   discussion towards this end is not provided but it would be very
   valuable for the readers.
4. Notation in the presentation of the method seems a bit
   inconsistent. I recommend authors to improve the consistency in the
   notation.

**Questions:**

1. can authors discuss further why the CIFAR experiments do not show a
   similar improvement?
2. can authors please improve the notation consistency in the
   presentation?

---

> ### Author Response · Authors · 2023-11-18
> **Author response**
>
> Thank you very much for your review, which encouraged us to further improve the paper.
>
> Weakness:
>
> 2. "-UA" uses the original prediction targets (as in Eq. 1), and the dilated prediction targets (the first half of Eq. 7) are not used. It uses multiple heads and $L_Q$ is still present. We have added more descriptions in the text.
>
> 3. We very much agree that this was a primary concern and we have run more experiments to check the reasons behind. We have added discussion on this and supplemented a small-scale experiment in Table 6 at the end of Appendix B. In general, we have used the same set of hyper-parameters from MSimCLR, but that may not be optimal. We noticed that a small $\tau$ or a linear increasing scheme of $\tau$ increases the performance of SimCLR-UA significantly.
>
> 4. We checked the paper carefully and spotted that the primary inconsistency in notations were caused by using $\theta$ to denote the NN encoder when formalizing DIM-UA. We thought about the pros and cons and decided to remove the use of $\theta$. We now use $f$ to denote the encoder and made changes based on that. Hopefully, this can increase the readability.
>
>     We have also added more descriptions to notations far apart from where they were first introduced.
>
> Questions:
>
> 1. Answered in Weakness 3.
>
> 2. Answered in Weakness 4.

---

> > ### Comment · Reviewer_Ww1x · 2023-11-22
> > **Thanks**
> >
> > Thanks for the clarification. I think with the proposed updates improve the quality of the article.

---

> ### Author Response · Authors · 2023-11-22
> **Kind reminder**
>
> Dear reviewer Ww1x,
>
> We really appreciate your input a lot and have improved our paper based on your advice since the discussion phase begined.
>
> Today is the last day of the discussion phase and we feel obliged to ask again if you have any further advice on this paper. We would like to discuss with you further to improve our paper.
>
> Thank you in advance.

---

### Official Review · Reviewer_ES5Q · 2023-11-01

**Soundness:** 3 good
**Presentation:** 2 fair
**Contribution:** 2 fair
**Rating:** 6
**Confidence:** 2

**Summary:**

The paper addresses the problem of learning a low-dimensional manifold from high-dimensional data in the context of state representation learning. A new approach based on self-supervised learning is proposed in order to learn an unbalanced atlas representation. The proposed approach is called DeepInfomax with an unbalanced atlas (DIM-UA) and is evaluated on 19 Atari games of the AtariARI benchmark. The evaluation shows good performance in this benchmark.

**Strengths:**

The paper addresses an important topic in the area of state representation learning. The proposed algorithm seems novel and yields good performance in the tested benchmarks.

**Weaknesses:**

I am struggling to understand the rationale behind the DIM-UA algorithm. What is the motivation for redefining the score function L_GL? Equation 9 has a hyperparameter for L_Q but not for L_GL or L_LL. Why? The writing of the paper is not clear in several points as some parts are difficult to follow. For example, Figure 1 is not quite clear to me and the caption is not very explicit. Additionally, the results could be presented in a more concise fashion (esp. Table 1 and 2 - showing the best results in bold would increase the readability).

**Questions:**

- What is the motivation for redefining the score function L_GL?

- Equation 9 has an hyperparameter for L_Q but not for L_GL or L_LL. Why?

- The improvements reported in table 2 seem relatively small. Can you comment on this further?

- Why is the linear evaluation accuracy on CIFAR10 a suitable evaluation metric? The improvements shown here (table 3) are very marginal at best.

---

> ### Author Response · Authors · 2023-11-18
> **Author response**
>
> Thank you very much for your kind review! We were sorry that the notations in DIM-UA were hard to follow. We put some effort on making the notations in DIM-UA more consistent. Hope it helps with the readability.
>
> Weakness:
> 1. We redefined $L_{GL}$ because the UA paradigm uses dilated prediction targets in pretraining, instead of using the normal prediction targets.
>
> 2. We have a hyper-parameter $\tau$ for $L_Q$ because this hyper-parameter controls the MMD loss on membership probabilities. For $L_{GL}$ or $L_{LL}$, they are directly controlled by the learning rate of the optimizer. We have showed that this hyper-parameter $\tau$ is indeed important. Please check Table 6 at the end of Appendix B to see the effects of changing $\tau$.
>
> 3. We have added more content to the caption of old Figure 1 (now Figure 2). We have made the highest score in each row bold, and also moved Table 2 to Appendix A (now Table 4). Those are really nice suggestions.
>
> Questions:
> 1. Answered in Weakness 1.
>
> 2. Answered in Weakness 2.
>
> 3. The old Table 2 now is labeled as Table 4, which has the accuracy scores. Our DIM-UA method has a mean accuracy of 76%, compared to 73% and 71% of ST-DIM using 256 units and using the same amount of units. First, VAE underperforms too much compared to ST-DIM, which may make a 3% increase seem not that big, but 3% increase in accuracy is quite decent since the difference between DIM-UA and ST-DIM is not even near to the difference between generative methods and contrastive methods.
>
>     Moreover, when comparing ST-DIM to DIM-UA with the same large number of units, DIM-UA wins by 5%, which suggests that a large number of hidden units in ST-DIM does not help with the performance. This is strong empirical evidence that our UA paradigm can create more effective representations.
>
> 4. This is a great point. We also thought that the performance of SimCLR-UA in Table 3 (now Table 2) needed more discussion. Our emphasis in the past few days has put on finding the key reason behind this. We have supplemented a small-scale experiment in Table 6 at the end of Appendix B. Mainly, we argued that the main reason why SimCLR-UA did not outperform SimCLR by a large margin is that it used the same set of hyper-parameters as in MSimCLR, which were not optimal. This, in turn, also suggests that SimCLR-UA beats MSimCLR by a large margin with no doubt.

---

> ### Author Response · Authors · 2023-11-22
> **Kind reminder**
>
> Dear reviewer ES5Q,
>
> Thank you very much for your constructive work again! We are reaching out because today is the last day of the discussion phase and we have not received any comment from you yet.
>
> Hope everything is fine! we look forward to discussing more about our paper with you.

---

### Official Review · Reviewer_hci6 · 2023-11-02

**Soundness:** 3 good
**Presentation:** 2 fair
**Contribution:** 3 good
**Rating:** 6
**Confidence:** 2

**Summary:**

This paper developed a state representation learning method leveraging an unbalanced atlas (UA). The authors have modified the ST-DIM algorithm to align with the proposed UA paradigm. Although the main contribution is not stated intuitively, empirical evaluations on 19 games of the AtariARI benchmark suggested an improved performance compared with three established baseline methods (many existing self-supervised methods are omitted for comparison). Furthermore, the authors performed a comprehensive ablation study for the design choices of the proposed method.

**Strengths:**

+ The experiments are conducted across 19 games of the AtariARI benchmark, covering a variety of vision tasks.

+ There are comprehensive ablation studies for the technical components of the proposed method.

**Weaknesses:**

- The clarity of the introduction could be enhanced by providing a more explicit context for the specialized terminology introduced (see Q1-3).

- The comparison would benefit from the inclusion of key baseline models which are currently absent (see Q4).

- Tables 1 and 2 appear to be redundant, presenting analogous results through different evaluative metrics (F1 score and Accuracy, respectively). Although a comprehensive evaluation is encouraged, putting these two sizable tables back to back in the main paper gives the impression of lacking sufficient materials for the paper. It would be more appropriate to consolidate these findings, perhaps through a combined analysis or in supplementary materials, to avoid repetition and maintain the conciseness of the paper.

- the paper lacks a clear statement of its underlying motivation and significance, which is pivotal for readers to comprehend the value and potential impact of the research (see Q5).

**Questions:**

1. The introduction used specialized terminology that may not be universally familiar, necessitating additional clarification for a broader audience. Specifically, the first sentence of the third paragraph introduces concepts such as *manifold*, *atlas*, *local structure*, and *chart*, which would benefit from further exposition to contextualize the study and its objectives.

2. The paper's motivation remains unclear, partly owing to the use of undefined terms. The concept of an *atlas*, and particularly the distinction between *unbalanced* and *balanced* atlases within this framework, needs clarification. The terms *prior distribution* and *membership probability distribution* introduced later also lack clear definitions, impeding the reader's understanding.

3. In the 3rd paragraph of Section 4, *d* and *n* are used without proper definition.

4. This paper suggests that pre-training a model using a reinforcement learning task and then fine-tuning it on downstream reinforcement learning tasks is beneficial. However, this point is not fully demonstrated because the authors did not compare the proposed method with other self-supervised learning methods, as reviewed in the introduction, e.g., contrastive models (SimCLR is compared) and generative models (none is compared).

5. From the introduction section, it is not intuitive to me why this study is important. For example, the last paragraph lists technical achievements but does not convey their broader significance. Specifically, (1) fitting the ST-DIM to UA paradigm (why UA paradigm is important?), (2) detailed ablations for better design choices (that's standard, not sure if it counts as a contribution), and (3) representing a manifold with a larger number (why this is important anyway?). Not limited to the introduction section, the authors did not describe the significance of the proposed method and all these ablation studies in the entire paper.

---

> ### Author Response · Authors · 2023-11-18
> **Author response**
>
> Thank you very much for your detailed review! We have moved Table 2 to Appendix A (now Table 4). Other part of weakness will be addressed in answering questions.
>
> Questions:
>
> 1. We agree that the terminology should be better introduced. We now have more descriptions in the caption of Figure 2.
>
> 2. The concept of an atlas is now described in Figure 2. Moreover, we described the notion of a uniform prior and unbalanced atlas in Introduction (above the contribution summary). We used the term 'balanced' in contrast to 'uniform'. While introducing these notions, we also strengthened our motivation towards our method. The membership probability can be viewed as a simple score function (by using Softmax) to attribute a probability score to each chart.
>
> 3. We have added short descriptions of $n$ and $d$ in Section 4.
>
> 4. We started our work with a focus on SRL, and we saw that generative methods did not perform well in SRL tasks (see the much lower F1 scores of VAE compared to CPC in Table 1). It was never our intention to omit generative methods, but we simply thought that contrastive methods were more suitable in SRL tasks. Of course, there are many other tasks where generative methods are more suitable, and it can be interesting future work.
>
> 5. This is a very good point. The reasons why this work is important should be better explained. Thus, we emphasized on our motivation in Introduction and extended the contribution summary. One important note is that our work can help build large models in deep learning by enlarging the heads (complementary to the typical research that deepens the backbones).

---

> > ### Comment · Reviewer_hci6 · 2023-11-22
> > **An annotated version might be helpful**
> >
> > Thank you for your response. I've briefly reviewed the revised manuscript and its accompanying response. To enhance the evaluation, I suggest marking the revised sections in the manuscript with color. This would be particularly useful as your response is concise and primarily directs the reader to the updated manuscript. Initially, I was unable to find the definitions for key terms like 'manifold', 'atlas', 'local structure', and 'chart' that I had inquired about. I will examine the manuscript more thoroughly, but my initial impression is that this paper may still need substantial revisions for a clearer presentation.
> >
> > Regarding the statement, *"our work can help build large models in deep learning by enlarging the heads (complementary to the typical research that deepens the backbones),"* I must admit that the motivation and significance of this approach are not entirely clear to me—for example, why is it important to 'enlarging the heads'?

---

> ### Author Response · Authors · 2023-11-22
> **Kind reminder**
>
> Dear reviewer hci6,
>
> We apologize for reaching out again. Today is the last day of the discussion phase and we have not heard from you yet.
>
> We have improved our paper a lot based on your valuable feedback. Please let us know if there is something more to be improved.

---

> ### Author Response · Authors · 2023-11-22
> **Thank you very much for your kind response**
>
> Dear reviewer hci6,
>
> Thank you very much for your response! Indeed, an annotated version would be very helpful. Please check our updated pdf. The terms are mainly introduced in the caption of Figure 2, since we thought that those terms can be better understood with a figure illustration.
>
> We agree that the term of local structure is still not independently introduced. However, please note that this term is not anything special in this context, it is more like a broad concept in maths. For instance, the concept of locality here is not much different compared to when we say that a function is locally smooth, or locally differentiable. It can be better understood to consider that the domain for a chart is local to the whole manifold for an atlas.
>
> For the motivation of enlarging heads, it is because we face bottlenecks when enlarging the backbone. For instance, over-parameterization is common for vision transformers (ViTs) and over-parameterized ViTs might perform worse on ImageNet 1K. When facing such bottlenecks, an alternative to build larger deep learning models would naturally be increasing the number of hidden units in output heads.

---

> > ### Author Response · Authors · 2023-11-23
> > **Only 2 hours left**
> >
> > Dear reviewer hci6,
> >
> > We updated our pdf and replied to your comment around 12 hours ago and are still awaiting your response. Did our last reply answer all your questions?
> >
> > As much as we would love to have a more detailed discussion with you, the update panel of the conference will close in 2 hours.
> >
> > We will strive to do our best to reply to your next comment provided that we could receive it before the end of the discussion phase.

---

### Author Response · Authors · 2023-11-18
**To all reviewers**

We would like to thank all reviewers for their great advice, which inspired us to improve our paper further.

We have updated our paper based on your advice (so we are referring to the updated labels in the current context).

First, we realized that the motivation of our paper might not be salient enough. We have moved related paragraph in Section 3.1 to Introduction (right above the summary of contribution). We also included Figure 1 of entropy to strengthen our motivation. We also extended our contribution summary.

We appreciate reviewers' concern regarding why SimCLR-UA could not beat SimCLR significantly on CIFAR10. We hypothesized that the main reason was that we used the hyper-parameters the same as MSimCLR without adaptations. After observing Figure 1, we were inspired to adjust the value of $\tau$ to control how fast the entropy drops. We found out that a small $\tau$ or a linear increasing schedule of $\tau$ can help improve the performance of SimCLR-UA significantly. Please check now the end of Appendix B for the detailed small-scale experiment on this.

Other revisions include adding more descriptions to sentences that were not clear to reviewers, and improving the notations in the paper. We also moved the old Table 2 (now Table 4) to Appendix A since it is very similar to Table 1.

---

### Meta-Review · Area_Chair_K7BF · 2023-12-18

**Metareview:**

The paper proposes to model different charts of an atlas via multiple heads of an encoder in the context of contrastive learning, specifically forcing a scoring/membership function to assign weights per chart that are different from uniform weighting. While the initial reviews raised concerns about the motivation for this work, from the AC's perspective, these concerns have been addressed satisfactorily during the rebuttal period. Considering the initial reviews, the rebuttal phase, and the discussion, I am recommending acceptance at this point.

**Justification For Why Not Higher Score:**

This paper presents a solid contribution, yet the scope is somewhat limited and might only be of interest to a sub-community. For those reasons, Accept (poster) appears to be a quite suitable choice.

**Justification For Why Not Lower Score:**

This is a solid contribution. There were some initial concerns, but those were eliminated (to a large extent) during the rebuttal, rendering this work a welcome contribution to ICLR.

---

### Decision · Program_Chairs · 2024-01-16

Accept (poster)